# Sorption/Diffusion Contributions to the Gas Permeation Properties of Bi-Soft Segment Polyurethane/Polycaprolactone Membranes for Membrane Blood Oxygenators

**DOI:** 10.3390/membranes10010008

**Published:** 2020-01-02

**Authors:** Tiago M. Eusébio, Ana Rita Martins, Gabriela Pon, Mónica Faria, Pedro Morgado, Moisés L. Pinto, Eduardo J. M. Filipe, Maria Norberta de Pinho

**Affiliations:** 1Department of Chemical Engineering, CeFEMA—Center of Physics and Engineering of Advanced Materials, Instituto Superior Técnico, Universidade de Lisboa, Av. Rovisco Pais, 1049-001 Lisbon, Portugal; tiago.eusebio@tecnico.ulisboa.pt (T.M.E.); anrivmartins@gmail.com (A.R.M.); gabriela_pon@hotmail.com (G.P.); monica.faria@tecnico.ulisboa.pt (M.F.); 2Department of Chemical Engineering, CQE—Centro de Química Estrutural, Instituto Superior Técnico, Universidade de Lisboa, Av. Rovisco Pais, 1049-001 Lisbon, Portugal; pm.elessar@gmail.com (P.M.); efilipe@tecnico.ulisboa.pt (E.J.M.F.); 3Departamento de Engenharia Química, CERENA, Instituto Superior Técnico, Universidade de Lisboa, 1049-001 Lisboa, Portugal; moises.pinto@tecnico.ulisboa.pt

**Keywords:** bi-soft segment polyurethane, gas permeation, solution-diffusion, time lag, integral asymmetric membranes, homogeneous symmetric membranes, membrane blood oxygenators

## Abstract

Due to their high hemocompatibility and gas permeation capacity, bi-soft segment polyurethane/polycaprolactone (PU/PCL) polymers are promising materials for use in membrane blood oxygenators. In this work, both nonporous symmetric and integral asymmetric PU/PCL membranes were synthesized, and the permeation properties of the atmospheric gases N_2_, O_2_, and CO_2_ through these membranes were experimentally determined using a new custom-built gas permeation apparatus. Permeate pressure vs. time curves were obtained at 37.0 °C and gas feed pressures up to 5 bar. Fluxes, permeances, and permeability coefficients were determined from the steady-state part of the curves, and the diffusion and sorption coefficients were estimated from the analysis of the transient state using the time-lag method. Independent measurements of the sorption coefficients of the three gases were performed, under equilibrium conditions, in order to validate the new setup and procedure. This work shows that the gas sorption in the PU/PCL polymers is the dominant factor for the permeation properties of the atmospheric gases in these membranes.

## 1. Introduction

Membrane blood oxygenators (MBOs), also known as artificial lungs, are medical devices that provide temporary partial or full support to patient lungs during cardiac surgeries where cardiopulmonary bypass (CPB) is performed, giving surgeons the possibility to operate in a motionless and bloodless environment. These devices are also used in longer duration therapies such as extracorporeal life support (ECLS), allowing the organs to heal from lung and/or cardiac failure [1,2]. During these medical procedures, the semipermeable membranes that compose the MBO assure the metabolic functions of the lung by providing efficient oxygenation and simultaneous depletion of carbon dioxide to and from the patients’ blood [3]. The two fundamental requirements for an MBO to be considered efficient are: (i) to perform efficient gas exchange by delivering approximately 250 cm^3^ (STP)/min of O_2_ and remove about 200 cm^3^ (STP)/min of CO_2_ at blood flow rates of 2–4 L/min; and (ii) be hemocompatible [4].

Despite 70 years of continuous developments and widespread use of MBOs, several aspects related to the membrane/blood interactions, circulating blood flow conditions, and equipment design still need to be improved [5,6]. The production of new and more efficient MBOs capable of promoting significant technological steps with clinical advantages is intrinsically connected to the development of novel gas exchange membranes. In the last years, research studies have focused on two main objectives: (i) the development of more hemocompatible materials; and (ii) the enhancement of the flow management/mass transfer associated to the metabolic functions of the lung [7,8]. This resulted in a wide variety of membranes made of different polymers either in the form of flat sheets or hollow fibers being incorporated in MBOs over the last decades. Further development of more efficient membranes is a major factor on the emergence of the next generation of MBOs which, by requiring less membrane area, will be smaller and will require lower priming volumes and lower blood flows.

Most of the polymer materials used in the past to fabricate membranes for MBOs, such as polypropylene (PP), polyethylene (PE), polymethylpentene (PMP), polyvinylidenefluoride (PVDF), and silicone rubber possess simple chemical structures as they are synthesized from one or two monomers, therefore leading to the formation of homopolymers or copolymers [3]. On the other hand, polyurethanes (PUs) possess more complex chemical structures that typically comprise three monomers: a diisocyanate, a polyol and a chain extender. Due to the three degrees of freedom that are thus available when designing the synthesis of a PU, one may obtain a virtually infinite number of materials with various physicochemical and mechanical characteristics. PU elastomers, which are segmented block copolymers, usually exhibit a two-phase structure in which hard segment (HS) enriched domains are dispersed in a matrix of soft segments (SS) [9,10]. Due to their structural versatility, physicochemical, and mechanical properties, as well as an enhanced bio/blood compatibility, PUs are the top candidate material for biomedical devices for cardiovascular applications (catheters, vascular prostheses, heart valves, etc.) and have established a niche in the medical device industry [11,12,13].

The introduction of a second type of SS further increases the versatility in the structure design of PU membranes due to the presence of a new chemical moiety, that can generate different degrees of phase separation between the two SSs and different degrees of phase segregation between the hard and soft segments. Bi-soft segment PU membranes containing poly(butadienediol) (PBDO) as a second SS were first synthesized by Zhao and de Pinho [14] for pervaporation membranes. Studies by Queiroz and de Pinho [15,16] showed that a membrane containing 20 wt% of PBDO showed phase separation of the two SSs and had higher CO_2_ permeability ranging from 150 to 950 Barrer. The membrane containing 67 wt% of PBDO which did not present phase segregation between the two SSs exhibited lower CO_2_ permeability ranging between 90 and 550 Barrer. 

The same group synthesized a second set of bi-soft segment PU membranes by introducing polydimethylsiloxane (PDMS) as the second SS [17]. The study revealed that the membranes with PDMS content ranging from 25 to 75 wt% showed phase separation of the two SSs and that the HSs formed small aggregates within these two phases. It was also concluded that the increase in PDMS content from 25 to 75 wt% led to the increase of permeabilities to CO_2_ from 200 to 800 Barrer and to O_2_ from 30 to 120 Barrer. Even though both the PU/PBDO and PU/PDMS membranes showed satisfactory gas permeation properties towards O_2_ and CO_2_, the blood compatibility properties were far from ideal [18]. This fact prompted the same research group to introduce polycaprolactone-diol (PCL-diol) as a second SS due to its recognized enhanced hemocompatibility properties and established use in vascular tissue engineering [19]. 

Nonporous symmetric bi-soft segment PU membranes containing PCL as the second SS were first synthesized by Besteiro et al. [20] by extending a triisocyanate polyurethane prepolymer with PCL-diol. Characterization studies showed that a variation of the PU/PCL weight ratio affects the surface energy, morphology, topography, and hemocompatibility. The nonporous symmetric PU/PCL membranes exhibited enhanced hemocompatibility properties in terms of being nonhemolytic, low thrombosis degree, as well as low platelet adhesion [20]. Studies of the chemical structure and phase segregation properties of the nonporous symmetric PU/PCL membranes revealed that the urethane groups form HS aggregates dispersed in the SS phase and that this aggregation increases with the increase of the PCL content [21]. Gas permeation properties of the nonporous symmetric PU/PCL membranes performed by a photo acoustic gas detection system revealed that the permeability of the nonporous symmetric PU/PCL membranes towards CO_2_ had a non-monotonous behavior, increasing from 188 to 337 Barrer when the PCL content increased from 0 to 10 wt%, but decreasing to 113 Barrer for the membrane containing 15 wt% of PCL. The O_2_ permeability was found to be independent of PCL content with values between 10 and 11 Barrer [21]. Despite the very promising results in terms of enhanced hemocompatibility, the gas permeability of the nonporous symmetric PU/PCL membranes was a shortcoming, not achieving the minimum O_2_ permeation requirements of an efficient MBO. In an attempt to increase the gas permeability of the PU/PCL membranes while at the same time preserving the hemocompatibility properties, the PU/PCL membranes were synthesized as integral asymmetric membranes with a very thin top dense layer and a thicker bottom porous support layer [22,23]. Extensive studies on the chemical composition, surface topography and blood compatibility of integral asymmetric PU/PCL membranes containing 5%, 10%, and 15% PCL brought great insight towards the tailoring of nanostructured dense layer surfaces with enhanced hemocompatibility [24,25]. Results showed that all of the membranes were non hemolytic, and that increase in PCL content was directly correlated to the minimization of platelet adhesion and non-promotion of extreme stages of platelet activation [24,25]. The volumetric permeation fluxes were measured for the single, pure gases, CO_2_ and O_2_, by the constant pressure method in a simple experimental setup described in previous work [17], and the results showed that the PU/PCL membrane containing 10 wt% PCL exhibited a CO_2_ permeance of 0.27 × 10^−5^ cm^3^(STP)/(cm^2^scmHg) which is above the required CO_2_ permeance for MBOs. In contrast, the O_2_ permeances were below the desirable threshold and further studies have to be carried out to determine the effect of the O_2_ solubility and of the O_2_ diffusion coefficients, in order to optimize the O_2_ permeance of the asymmetric PU/PCL membranes [22]. 

The present work addresses this problem of decoupling the solubility/diffusion contributions to the N_2_, O_2_ and CO_2_ gas permeance. For that effect, a new high precision gas permeation measurement apparatus was designed, built, tested and validated. The built system is a barometric, constant volume device that, at constant temperature and feed gas pressure, records the evolution of the permeate pressure online. It allows the precise measurement of both the transient and steady state permeation regimes for a wide range of permeances and consequently of the gas permeability, diffusion and solubility coefficients in the studied membranes. It is here used in the characterization and optimization of bi-soft segment PU/PCL membranes destined for future MBOs.

First, four nonporous symmetric PU/PCL membranes containing 0, 5, 10, and 15 wt% of PCL, were synthesized by the solvent evaporation method in order to determine the effect of the second soft segment (PCL) content on the gas permeation properties of the PU/PCL membranes.

In a second group of membranes, three types of integral asymmetric membranes containing 5, 10 and 15 wt% PCL were synthesized by the phase inversion techniques, with solvent evaporation times of 5 and 10 min for each casting composition.

The surface and cross section structures of these membranes were characterized by scanning electron microscopy (SEM) and permeate pressure vs. time curves were obtained for the three gases at several gas feed pressures. The steady-state fluxes were determined and the diffusion (*D*) and solubility (*S*) coefficients were obtained from the transient state part of the curves by the time-lag method. In order to validate these results, equilibrium sorption isotherms were measured for each pure gas in a test membrane, using an independent setup, and the results are compared to the time-lag obtained values.

## 2. Theory

### 2.1. Solution–Diffusion Model

The transport of a single gas through a dense, nonporous polymeric membrane can be described by the solution-diffusion model [26]. Being the pressure difference across the membrane the driving force, the gas molecules dissolve in the upstream face of the membrane, diffuse across the membrane and desorb from the downstream face of the membrane. In the steady-state, the unidimensional diffusive flux is described by the Fick´s First Law of diffusion:(1)$$J_{A}=-D_{A}\frac{dC_{A}}{dx}$$
where *J_A_* is the flux of species *A* in the *x* direction and is proportional to the concentration gradient $\frac{dC_{A}}{dx}$, $C_{A}$, is the concentration of species *A* in the membrane and $D_{A}\;$ is a proportionality constant defined as the diffusion coefficient, independent of the solute *A* concentration in this low concentrations range.

Integrating Equation (1) over the thickness of the membrane, *l*, gives:(2)$$J_{A}=\frac{D_{A}}{l}\left(C_{A0}-C_{Al}\right)$$
where $C_{A0}$ and $C_{Al}$ are the concentrations of *A* in the membrane on the feed side, and permeate side, respectively.

The solubility of gases in elastomers is usually very low and can be described by Henry´s Law, given by Equation (3), where the concentration inside the polymer, C, is proportional to the applied pressure, p.
(3)C=Sp

By applying Henry´s Law, the following relations can be established:(4)$$S_{A}=\frac{C_{A0}}{p_{f}}=\frac{C_{Al}}{p_{p}}$$
where *S_A_* is the solubility coefficient of *A*, *p_f_* is the pressure of the feed and *p_p_* is the pressure of the permeate [27].

Combining Equations (2) and (4) gives the following expression:(5)$$J_{A}=\frac{D_{A}S_{A}}{l}\left(p_{f}-p_{p}\right)$$

Making the product *D_A_S_A_* equal to the permeability coefficient, *P_A_*, Equation (5) can be written as:(6)$$J_{A}=\frac{P_{A}}{l}\left(p_{f}-p_{p}\right)$$

When the thickness is difficult to define, the pressure normalized flux or permeance, *P_A_*/*l*, is used instead:(7)$$Perm=\frac{P_{A}}{l}$$

### 2.2. Diffusion in Transient State: Time–Lag Method

In the transient state, the mass balance of the unidimensional diffusive transport of species A through a dense, nonporous polymeric membrane is given by the following expression:(8)$$-\frac{dC_{A}}{dt}=\frac{dJ_{A}}{dx}$$

Substituting the flux by the Fick’s First Law Equation (1), the Fick’s Second Law is obtained:(9)$$\frac{dC_{A}}{dt}=D_{A}\frac{d^{2}C_{A}}{dx^{2}}$$

If the membrane is initially free of the diffusing species, the following initial and boundary conditions for the system can be applied:(10)$$C_{A}\left(x,0\right)=0$$
(11)$$C_{A}\left(0,t\right)=C_{A0}$$
(12)$$C_{A}\left(l,t\right)=C_{Al}\approx 0$$
which means that the upstream concentration, *C_A_*_0_, remains constant and the downstream concentration, *C_A_*_l_, is negligible compared to the upstream during the diffusion process. Fulfilling these boundary conditions, the solution of Equation (9), either by Laplace transform or separation of variables, is given by [28,29]:(13)$$C_{A}=C_{A0}\left(1-\frac{x}{l}\right)+\frac{2C_{A0}}{l}\times \overset{\infty }{\underset{n=1}{\sum }}\frac{1}{n}\sin\left(\frac{n\pi x}{l}\right)exp\left(-\frac{D_{A}n^{2}\pi^{2}t}{l^{2}}\right)$$

The solution expressed in terms of the diffusive flux can be obtained by substituting Equation (13) in the Fick’s First Law:(14)$$J_{A}\left(x,t\right)=\frac{D_{A}C_{A0}}{l}+\frac{2D_{A}C_{A0}}{l}\times \overset{\infty }{\underset{n=1}{\sum }}\cos\left(\frac{n\pi x}{l}\right)exp\left(-\frac{D_{A}n^{2}\pi^{2}t}{l^{2}}\right)$$

The first term in Equation (14) is the steady state portion of the flux and the second term represents the transient contribution. It is a function of time and displacement in the direction of diffusion and hence can be solved for the fluxes entering and leaving the membrane (*x* = 0 and *x* = l, respectively).

By setting *x = l*, yields a time-dependent flux equation relative to the downstream end of the membrane. Integrating it with respect to time, yields the amount of species *A* permeating out of the membrane, *Q_Al_*:(15)$$Q_{Al}\left(t\right)=-A\int_{0}^{t}J_{A}\left(t\right)dt=\frac{AD_{A}C_{A0}}{l}\left[t-\frac{l^{2}}{6D_{A}}+\frac{2l^{2}}{\pi^{2}D_{A}}\times \overset{\infty }{\underset{n=1}{\sum }}\frac{{\left(-1\right)}^{n+1}}{n^{2}}exp\left(-\frac{D_{A}n^{2}\pi^{2}t}{l^{2}}\right)\right]$$

The permeate pressure is, then, obtained from the amount of species *A* permeating out of the membrane:(16)$$p_{p}\left(t\right)=\frac{AD_{A}p_{f}}{Vl}\left[t-\frac{l^{2}}{6D_{A}}+\frac{2l^{2}}{\pi^{2}D_{A}}\times \overset{\infty }{\underset{n=1}{\sum }}\frac{{\left(-1\right)}^{n+1}}{n^{2}}exp\left(-\frac{D_{A}n^{2}\pi^{2}t}{l^{2}}\right)\right]$$
where *A* is the cross-sectional area available for gas penetration perpendicular to the direction of diffusion and *V* is the volume of the receiving chamber. The steady-state asymptote of Equation (16) is found by taking the limit as *t*→∞, reducing the transient term to zero. The permeate pressure is, then, given by:(17)$$\underset{t\to \infty }{\lim}p_{p}\left(t\right)=\frac{AD_{A}p_{f}}{Vl}\left[t-\frac{l^{2}}{6D_{A}}\right]$$

The intercept on the time axis of the plot of pressure rise versus time is defined as the time lag, *t_lag_*:(18)$$t_{lag}=\frac{l^{2}}{6D_{A}}$$

Thus, from the time lag and knowing the membrane thickness, the diffusion coefficient can be obtained.

## 3. Materials and Methods

### 3.1. Materials

Two prepolymers were used for the synthesis of the polyurethane membranes: (i) a poly(propylene oxide) (PPO) based polyurethane prepolymer with three isocyanate terminal groups (PUR), supplied by Fabrires-Produtos Químicos S.A. (Lisbon, Portugal) and (ii) a polycaprolactone diol prepolymer (PCL-diol), supplied by Sigma-Aldrich (Lisbon, Portugal) with a stated molecular weight of 530 Da. The solvents used were dimethylformamide (DMF) (p.a. grade, 99.8%) and diethyl ether (DEE) (p.a. grade, 99.5%) provided by Panreac (Barcelona, Spain). The catalyst used was Tin(II) 2-etilhexanoate (p.a. grade, 95%) provided by Sigma-Aldrich (Lisbon, Portugal).

Gas permeation experiments and gas solubility measurements were carried out using nitrogen (purity ≥ 99.999%), carbon dioxide (purity ≥ 99.98%) and oxygen (purity ≥ 99.5%), all supplied by Air Liquide (Lisbon, Portugal).

### 3.2. Membrane Synthesis

Group 1: Four casting solutions with a total prepolymers:solvent weight ratio of 65:35, a solvent ratio of DMF/DEE of 3:1, and containing different proportions of the two pre-polymers PU and PCL were prepared and subjected to the reaction conditions as previously described [21,30]. Membrane preparation was concluded by spreading each casting solution on a glass plate with a 250 μm Gardner knife and exposed to the atmosphere for 24 h. The four resulting membranes differed in the relative amount of PCL: 0%, 5%, 10%, and 15%, which were designated by PU0, PU5, PU10, and PU15, respectively.

Group 2: Three sets of integral asymmetric PU/PCL membranes containing 5%, 10%, and 15% of PCL, were synthesized by a modified version of the phase inversion technique where, as above described, the casting solutions with the PU and PCL pre-polymers were subjected to reaction conditions at room temperature and stirring for 2 h [22,24]. For each proportion of PCL, two membranes were prepared following exactly the same procedure as for group 1, up to the step of spreading the casting solution on the glass plates. At this point, instead of letting the membranes to dry completely, solvent evaporation times of 5 or 10 min were applied, after which the glass plates were placed in a coagulation bath of deionized water for at least 12 h. These membranes were then removed from the glass plate, washed with deionized water to remove all traces of solvent and left to dry at room temperature. These membranes were named according to the convention PU*x-y*, where *x* is the weight percentage of PCL and y is the solvent evaporation time in min; the membranes prepared were thus denominated PU5-5, PU5-10, PU10-5, PU10-10, PU15-5, and PU15-10.

The synthesized membranes were stored at ambient temperature, exposed to the air, and all the characterization and measurement procedures described below were performed during the period of about one month after the synthesis. No alteration of the properties of the membranes was detected during this time.

### 3.3. Scanning Electron Microscopy

Both the PU/PCL membranes from group 1 (nonporous symmetric) and group 2 (integral asymmetric) were characterized by scanning electron microscopy (SEM), using a JSM-7001F FEG-SEM microscope (JEOL, Tokyo, Japan). The samples were fractured after freezing in liquid nitrogen, mounted on a stub, and sputter-coated with gold. Images of the top and bottom surfaces and of the cross-sections were obtained.

### 3.4. Gas Permeation by the Constant Volume Method

#### 3.4.1. Experimental Setup

The gas permeation properties of the PU membranes were determined by the constant volume method using the experimental setup schematized in Figure 1. This method consists in applying a constant pressure of gas on the feed side of the membrane and then following the gas flux through the membrane by measuring the variation of pressure with time in a constant volume receiving chamber on the permeate side.

The permeation cell [30] is a flat plate cell with two detachable parts separated by a porous plate (membrane support). The effective membrane surface area is 9.62 cm^2^. The feed side of the cell is connected through a valve (V1) to the pressure regulator valves (PRV) of the feed gas cylinders and to the Setra model 205-2 (Boxborough, MA, USA) feed pressure manometer (PfT). The permeate side is connected to two cylinder buffers of different sizes (12.6 ± 0.1 cm^3^ and 167.2 ± 0.2 cm^3^), each with its respective valve (V6 and V7), and to the Paros model 6100A-CE (Redmond, WA, USA) permeate pressure manometer (PpT). Both sides of the setup can be opened to a vacuum pump (Edwards model E2M2 (Burgess Hill, UK), *p* < 0.1 mbar) or to the atmosphere (through V2–V5). The connections between the parts are made of stainless steel 316 1/8 inch O.D. tubing with the respective tube fittings (Gyrolok^®^), and the needle valves used are Hoke^®^ 3700 series. The equipment is installed inside a thermostatic air bath, which consists of a glass door fridge (wine cellar) that acts as the insulated box and cold source, a Hart Scientific PID controller connected to a platinum resistance thermometer and a heater, and two fans to homogenize the inner temperature.

It should be mentioned that, during the building and testing phases of this new apparatus, the configuration of permeate side had to be optimized by choosing the best relative placements of the PpT manometer and of the buffer cylinders, as well as the respective volumes. This was done to minimize the non-negligible resistance to gas transport observed during measurements as a consequence of Knudsen flow at the lowest pressures, which affected the obtained results. The final configuration, shown in Figure 1, was optimized following the studies and recommendations of Kruczek and coworkers [[31][32],[33],[34]], and thorough tests showed that the effect of Knudsen flow is non-detectable in the present configuration. 

The volume of the permeate side receiving chamber can be chosen by manipulating valves V6 and V7, according to the permeance of the measured gas through the membrane sample, to maximize the accuracy of the measurements. The permeate pressure is automatically recorded on a computer as a function of time (Digiquartz^®^ version 2.0 software, Paroscientific Inc., Redmond, WA, USA). 

#### 3.4.2. Procedure

Each series of measurements started by inserting the membrane sample in the permeation cell and thermostatizing the experimental setup until a stable temperature of 37.0 ± 0.2 °C was achieved. Prior to initiating the permeation experiment, the membrane is degassed using the vacuum pump, with valves V1 and V5 closed and all other valves opened. For each pure gas (N_2_, O_2_, or CO_2_), a series of measurements was done by regulating the respective pressure reducing valve (PRV) to a feed pressure between 0 and 5 bar. Then, to start the measurement, valves V2 and V3 are closed and the permeate pressure recording starts when V1 is opened. The feed pressure manometer is monitored during the measurement to ensure that the feed pressure is constant. After the measurement, V1 is closed and V2 and V3 are opened to degas the setup again with the vacuum pump before the next measurement. It was found that a degassing time of 10 min sufficed to completely remove any gas in the membrane as no increase in pressure was observed after closing valve V3.

### 3.5. Gas Solubility by the Barometric Method

A second experimental setup, schematically presented in Figure 2, based on the barometric method, was used to independently measure the solubility coefficients of the three studied gases (N_2_, O_2_, and CO_2_) in a test membrane. This was done to further validate the estimates of the solubility (and hence also diffusion) coefficients obtained by the application of the time-lag method to the pressure vs. time results obtained with the newly built gas permeation setup. A detailed description of the working principles of this gas solubility apparatus and of a similar experimental setup has been previously reported [35]. The test membrane (0.3147 g sample) was prepared according to the procedure described above (Section 3.2, Group 2), using a 10% proportion of PCL, 1 min of solvent evaporation time and a 150 µm casting knife. It was inserted in a cell, placed in the apparatus and put under high vacuum with a turbomolecular pumping station (HiCube 80, Pfeiffer Vacuum, Aßlar, Germany) capable of vacuum lower than 10^−2^ Pa, for at least an hour at 50 °C. The apparatus has one pressure transducer (MKS Instruments, Baratron 627D14TBC, Andover, MA, USA) and two chambers that are separated by a valve: one which volume was previously calibrated (chamber 1) and another where the cell containing the membrane sample is inserted (chamber 2). The volume available for the gas in chamber 2 was calibrated with helium when the membrane was already inside. The temperature of both chambers was controlled at 37.5 ± 0.01 °C using a water bath (Julabo MB, Seelbach, Germany). The gas is introduced in chamber 1 at a desired pressure. The gas is then expanded to chamber 2 by opening the valve and the pressure decay is monitored until it reaches equilibrium (a constant value is reached after 30–60 min). The final pressure value is registered and the sorbed gas is determined by mass balance. More gas is introduced into the system for the next measurement and this process is repeated until a complete isotherm is obtained. To calculate the sorbed amounts, the non-ideality of the gas phase is considered using the second and third virial coefficients [36].

To prevent contamination of the volumetric apparatus by the solvents used in the membrane synthesis, the membrane was previously dried under vacuum (*p* < 10 Pa) in a Schlenk flask at ambient temperature, for at least a week.

### 3.6. Statistical Analysis

To compare the different mean average values, a one-way analysis of variance (one-way ANOVA, *p* < 0.05) was used, using Tukey’s HSD (honestly significant differences) test to identify the ones that are significantly different. A two-tailed Student’s *t*-test was used to compare two mean average values after an F-test to determine if the two variances were equal.

## 4. Results and Discussion

### 4.1. SEM

Figure 3 shows the SEM images of the top and bottom surfaces and cross-section of the nonporous symmetric PU5 membrane. The micrographs show that the PU5 membrane is completely dense with no detectable pores. The SEM images obtained for all of the other nonporous symmetric membranes of group 1, PU0, PU10 and PU15, shown in Figure A1 of Appendix A, are similar to the ones obtained for the PU5 membrane having no visible pores. As expected, the PCL content seems to have no effect on the morphology of the nonporous symmetric membranes, since membranes were prepared by the solvent evaporation method where the membranes are left to dry for at least 24 h during which both of the solvents completely evaporate. 

SEM images were also obtained for the integral asymmetric membranes from group 2. The micrographs of the top surface, bottom surface, and cross-section of the PU15-5 and PU15-10 membranes are shown in Figure 4. Observing the top surface views in Figure 4a,d relative to membranes PU15-5 and PU15-10 and comparing with the bottom surfaces in Figure 4b,e, both membranes show larger and more numerous pores on the bottom surfaces, distinctly different from the top ones. This difference in number and size of the observed pores confers the asymmetric cross-section structures seen in Figure 4c,f.

SEM images obtained for the rest of the membranes of group 2, PU5-5, PU5-10, PU10-5, and PU10-10 have very similar features to the ones described for the PU15-5 and PU15-10 membranes and are shown in Figure A2 and Figure A3 of Appendix A.

The total thickness (l) of the studied membranes was measured with a digital caliper on four different regions of each sample and confirmed with measurements performed on five points of the cross-section SEM micrographs of each membrane using the ImageJ software [37]. Table 1 shows the average values obtained for each membrane, along with the respective standard deviations.

### 4.2. Permeation of CO_2_, O_2_, and N_2_ through Nonporous Symmetric PU Membranes (Group 1)

For each single pure gas permeating through a 9.62 cm^2^ membrane, the variation of the permeate pressure into the receiving chamber (initially at vacuum state) is recorded as a function of time. The experiments were performed at a temperature of 37.0 ± 0.2 °C, for feed pressures between 112 and 465 cmHg. Figure 5 shows an example of N_2_, O_2_ and CO_2_ permeation curves, with the steady state asymptote, corresponding to a permeation experiment of each single pure gas through the PU10 membrane.

It can be seen in Figure 5 that, after 100 s, the permeate pressure increased 0.13, 0.30, and 2.91 cmHg for N_2_, O_2_, and CO_2_, respectively, and that the slope of the steady state region of the curve increases in the order of N_2_, O_2_, and CO_2_. These pressure values are typical for the membranes studied. For the initial permeation times (<20 s) a transient state can also be identified, and this region has been amplified in the inset of Figure 5.

Permeation curves of N_2_, O_2_ and CO_2_ were obtained at several feed pressures (*Pf*) for all of the studied PU/PCL membranes. The slope, (*dp_p_*)/*dt*, of the steady state region of each of the permeation curves was transformed into molar flow, *dn*/*dt*, using the Ideal Gas Law:(19)$$\frac{dn}{dt}=\frac{dp_{p}}{dt}\frac{V_{s}}{RT}$$
where *Vs* is the volume of the receiving chamber and *T* is the absolute temperature at which the experiments are performed. It was verified that the corrections due to gas non-ideality [36] were negligible. Then, the molar flow is converted into volumetric flow, $\frac{dV_{STP}}{dt}$, in STP conditions:(20)$$\frac{dV_{STP}}{dt}=\frac{dn}{dt}\frac{RT_{STP}}{P_{STP}}$$
where *V_STP_*, *T_STP_*, and *P_STP_* are the volume, temperature, and pressure in STP conditions. Substituting Equation (19) in Equation (20), the following expression is obtained:(21)$$\frac{dV_{STP}}{dt}=\frac{dp_{p}}{dt}\frac{V_{s}T_{STP}}{TP_{STP}}$$

The volumetric flux is then obtained by dividing the volumetric flow by the effective membrane area, *A* (9.62 cm^2^):(22)$$J=\frac{dV_{STP}}{Adt}$$

The permeance of each single pure gas, *Perm*, is defined by: (23)$$Perm=\frac{dJ}{d\left(TMP\right)}\;\left[\frac{cm^{3}\left(STP\right)}{cm^{2}\;s\;cmHg}\right]$$

The average permeability coefficient (*P*) towards each of the three gases is determined by:(24)$$P=Perm\;\times \;l\;\times \;10^{10}\;\left[Barrer\right]$$
where *l* is the membrane thickness.

Figure 6 shows the volumetric flux (*J*) of N_2_, O_2_, and CO_2_ as a function of the transmembrane pressure (*TMP*), which is defined as the difference between the feed pressure and the initial permeate pressure, for the PU0, PU5, PU10, and PU15 dense membranes (group 1). As can be seen, the steady-state volumetric fluxes *J* increase linearly as a function of TMP in each series of measurements. Moreover, the values of *J* for CO_2_ are one order of magnitude higher than for the other two gases, and higher for O_2_ than for N_2_. For all the gases, higher fluxes were observed through the membrane without PCL (PU0), and then decreased in a non-monotonous fashion with the PCL proportion of the polymer; the lowest fluxes were always measured for PU15, whereas PU5 and PU10 presented similar fluxes, within the experimental uncertainty.

Table 2 shows the permeances, obtained from the slopes of the J vs. TMP plots depicted in Figure 6, and the permeability coefficients (P), obtained by Equation (24) using the values of membrane thickness l in Table 1, towards N_2_, O_2_ and CO_2_ for the PU0, PU5, PU10, and PU15 nonporous symmetric membranes.

The results show that both the CO_2_ permeances and permeability coefficients are approximately 25 times and 10 times higher than for N_2_ and O_2_, respectively. As for the fluxes, both the permeance and permeability values of all gases decrease with the PCL content of the polymer.

The permeability coefficients *P* show the same tendency, with the values of *P* for CO_2_ being approximately 30 times and 10 times higher than those for N_2_ and O_2_, respectively. 

Faria et al. [21] reported *P*(CO_2_) values between 113 and 337 Barrer and *P*(O_2_) values between 10 and 11 Barrer for nonporous symmetric membranes containing 0–15 wt% PCL, using a different measuring technique and a different PU prepolymer. For the nonporous symmetric membrane containing 10 wt% of PCL, the values of *P*(CO_2_) and *P*(O_2_) were 337 Barrer and 11 Barrer, respectively, which is higher than the *P*(CO_2_) (193 Barrer) and lower than the *P*(O_2_) (20 Barrer) values obtained in this work for the PU10 membrane.

When compared to permeability coefficients of membranes used in current MBOs, the *P*(CO_2_) values obtained for the PU membranes are higher than the ones claimed for the PP (9 Barrer) and PMP (90 Barrer) membranes. The *P*(O_2_) values obtained for the PU membranes are in between the values claimed for the PP (2 Barrer) and PMP (30 Barrer) membranes [38]. These values, allied to an enhanced hemocompatibility, confirm the potential of the PU/PCL polymers for use in membranes for blood oxygenators.

### 4.3. Determination of the CO_2_, O_2_, and N_2_ Diffusion and Solubility Coefficients of the Nonporous Symmetric PU Membranes by the Time-Lag Method

The diffusion (*D*) and solubility (*S*) coefficients of each gas in the PU0, PU5, PU10, and PU15 membranes were estimated from the permeation curves (*Pp* vs. time) by the time lag method. An example of application of this method is shown in Figure 7 (for the PU10 membrane at a feed CO_2_ pressure of 269 cmHg), which illustrates the determination of the time lag (*t_lag_*) as the intersection of the steady state asymptote with the x axis (time). The diffusion coefficient (*D*) is calculated using Equation (18), and the solubility coefficient (*S*), is then obtained from the product *P = DS* (from Equation (5)), using the values of the permeability coefficients (*P*) determined by Equation (24) and presented in Table 2.

Table 3 shows the values of *t_lag_*, *D* and *S* obtained towards N_2_, O_2_, and CO_2_ for the nonporous symmetric PU0, PU5, PU10, and PU15 membranes.

Regarding the diffusion coefficients D, the values found are always in the order *D*(N_2_) < *D*(CO_2_) < D(O_2_) for all membrane compositions. The order of the N_2_ and O_2_ diffusion coefficients may be related to their kinetic diameters, as seen in Table 4, with the smaller diameter of O_2_ imparting a higher mobility and, hence, a higher D. The intermediate value for CO_2_, despite its smaller effective diameter, may be due to its polar character, that promotes polar interactions with the polymer matrix of the membranes which may hamper its mobility. For each gas, the diffusion coefficient seems to decrease with the PCL percentage of the membrane. 

The solubility coefficients S, in a given membrane, are always much larger for CO_2_ than for the other two gases, and higher for O_2_ than for N_2_. It has been proposed that the solubility is related to the gas boiling point or critical temperature [39]. As shown in Table 4, CO_2_ has the highest boiling point, which correlates to its high solubility in the membrane; the boiling points increase in the order N_2_ < O_2_ < CO_2_, and this same trend is observed in the obtained solubility coefficients shown in Table 3. The solubility of each gas as a function of PCL content is not monotonous, with an apparent general tendency of S decreasing with PCL percentage, but with the PU10 membrane showing the highest solubility coefficients for all gases.

Faria et al. [21] also obtained D_CO2_ and S _CO2_ values for a membrane containing 10 wt% of PCL but a PU prepolymer different from that used in this work. The values obtained, 8.15 × 10^−7^ cm^2^/s and 4.14 × 10^−2^ cm^3^/cm^3^cmHg, respectively, are of the same order of magnitude as the values obtained in this study.

A joint analysis of the coefficients in Table 2 and Table 3 clearly shows that the permeation of the different gases through the PU/PCL polymeric membranes studied is a solubility controlled process. Although the three gases present slightly different diffusion coefficients, it is the disparity in their solubility coefficients that induces the large differences observed in the permeabilities.

### 4.4. CO_2_, O_2_, and N_2_ Permeation of the Integral Asymmetric PU Membranes from Group 2

Figure 8 shows the volumetric flux (*J*) of N_2_, O_2_, and CO_2_ vs. TMP for the integral asymmetric membranes from group 2. The respective permeance results, calculated from the slopes of the linear fits, are presented in Table 5. These membranes display an asymmetric cross-section consisting of two regions, one dense and one porous. However, it is not trivial to measure the thickness of the active (dense) layer and, therefore, we have chosen to show only the values of permeance in Table 5.

As can be seen, the relative fluxes and permeances of the three gases through each membrane follow the behavior observed for the dense membranes of group 1, with CO_2_ presenting permeances one order of magnitude higher than the other two gases, being around 10 times higher than O_2_ and 30 times higher than N_2_. For the membranes with the same evaporation time, a slight tendency of decreasing permeability with increasing PCL content can be identified, as was observed for the dense membranes. The membranes prepared with the shorter evaporation time (5 min) are more permeable to all the three studied gases than those with 10 min of evaporation. However, these asymmetric membranes present permeance values which are very close to those observed for the dense membranes of group 1. Further studies are needed in order to better understand the structure of the porous and dense layers and how to decrease the resistance of these membranes to gas transport.

### 4.5. Determination of the CO_2_, O_2_, and N_2_ Solubility Coefficient for a Test Membrane by the Barometric Method

In order to further validate the solubility coefficient estimates obtained from the application of the time-lag method, independent measurements of the solubility of the three studied gases were performed by the barometric method on a test membrane, on the previously described sorption apparatus (Section 3.5), and compared with the estimates obtained for the same membrane using the time-lag method. The test membrane used was prepared according to the procedure described above (Section 3.2, Group 2), using a 10% proportion of PCL, 1 min of solvent evaporation time and a 150 µm casting knife. The solubility measurements were performed on a 0.3147 g sample of membrane. Sorption isotherms at 37.50 ± 0.01 °C were obtained for the three studied gases as a function of pressure up to 300 cmHg, and the results are shown in Figure 9.

The solubility coefficients obtained from the slope of each isotherm, assuming the validity of Henry’s Law, were of (9.0 ± 1.7) × 10^−4^ cm^3^/cm^3^cmHg for N_2_, (27.2 ± 2.6) × 10^−4^ cm^3^/cm^3^cmHg for O_2_, and (198.0 ± 0.6) × 10^−4^ cm^3^/cm^3^cmHg for CO_2_. The higher uncertainties obtained for the O_2_ and N_2_ coefficients are due to the very low solubility of these gases in the membrane, which result in sorbed amounts of gas which are close to the limit of application of the used setup. The results are compared in Figure 10 with the S values obtained by the time lag method for the same membrane, where it can be seen that the solubilities obtained by the barometric method coincide within the experimental uncertainty with the results obtained from the time lag method for all three gases, fully validating the application of this method to the pressure vs. time data, and the solubility and diffusion coefficients obtained therefrom.

## 5. Conclusions

The gas permeation properties of biocompatible bi-soft segment polyurethane/polycaprolactone membranes to the respiratory gases N_2_, O_2_, and CO_2_ were studied in this work.

To this effect, a new constant volume gas permeation setup was built, that allows the precise measurement of the permeation pressure vs. time curve for a single gas through a membrane, for a given constant feed pressure. The equipment has a permeation cell for a 9.62 cm^2^ membrane and is able to work at constant temperatures from 10 to 50 ± 0.2 °C, with feed pressures up to 8 bar. The obtained pressure vs. time curves allow the determination of the steady state gas flux and, thus, the membrane permeance, and also the application of the time lag method to estimate the diffusion and solubility coefficients. Independent measurements of the solubility coefficient of the three studied gases in a test membrane were made in a previously built and validated setup, and the obtained results agree with the time lag measurements within the experimental uncertainty.

Nonporous symmetric PU/PCL membranes were synthesized and cast by the solvent evaporation technique, with PCL proportions of 0, 5, 10, and 15 wt%, and an average thickness of around 110 µm. The measured permeability coefficients for CO_2_ (between 172 and 227 Barrer) were around 10 and 25 times higher than those for O_2_ (17–24 Barrer) and N_2_ (7–9 Barrer), respectively. The permeation of the three gases slightly lowers when the proportion of PCL in the membrane increases, but the measured permeability coefficients compare favorably with the polymers currently used in commercial membrane blood oxygenators. The decoupling of the diffusion and solubility coefficients obtained by the time-lag method shows that the permeation process of the respiratory gases through PU/PCL membranes is controlled by solubility.

Integral asymmetric membranes were also produced, using the phase inversion technique, with 5, 10 and 15 wt% proportions of PCL and both 5 and 10 min of evaporation time. The 5 min membranes presented higher permeabilities when compared to the 10 min ones, but no enhancement was observed when comparing to the nonporous membranes. Further studies are needed to optimize the morphology of the membrane and the thickness of the dense layer.

## Figures and Tables

**Figure 1 membranes-10-00008-f001:**
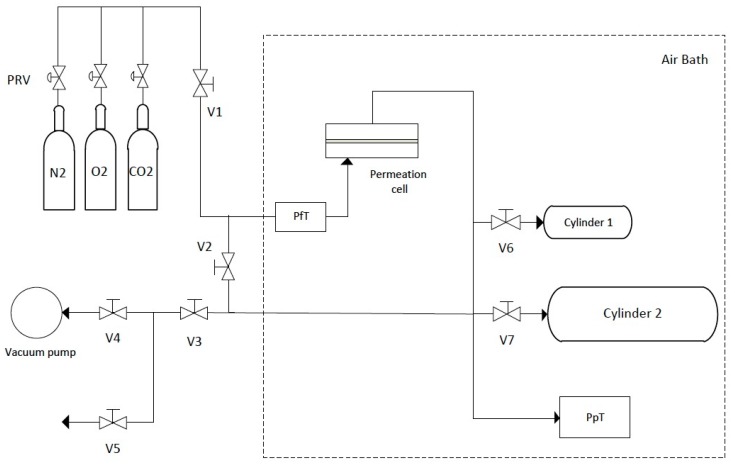
Schematic representation of the custom-built experimental gas permeation setup. V1–7: needle valves, PRV: pressure regulator valves, PfT: feed pressure manometer, PpT: permeate pressure manometer.

**Figure 2 membranes-10-00008-f002:**
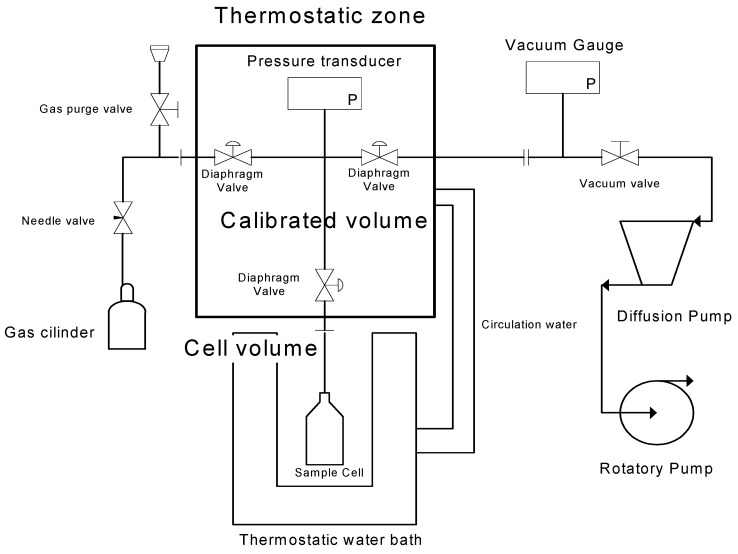
Schematic representation of the barometric apparatus for gas sorption experiments.

**Figure 3 membranes-10-00008-f003:**
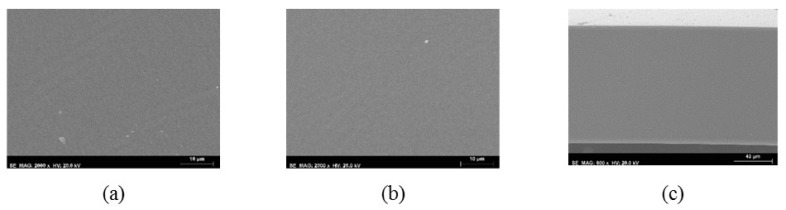
SEM images of the nonporous symmetric PU5 from group 1: (**a**) top surface (2000×), (**b**) bottom surface (2000×), (**c**) cross-section (600×).

**Figure 4 membranes-10-00008-f004:**
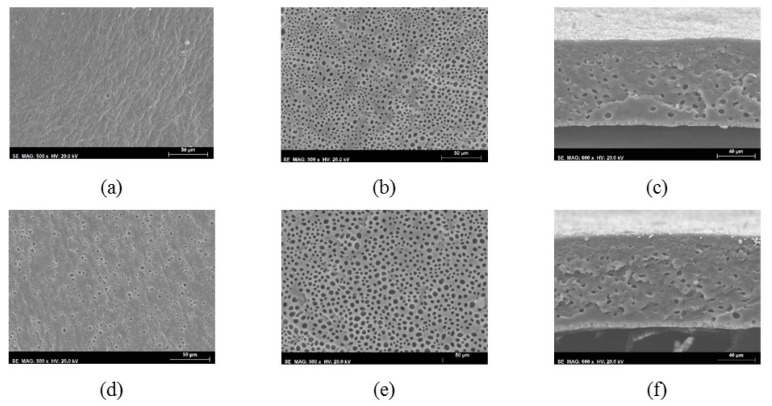
SEM images of the integral asymmetric PU15-5 and PU15-10 membranes: (**a**) top surface of PU15-5, (**b**) bottom surface of PU15-5, (**c**) cross-section of the of PU15-5; (**d**) top surface of PU15-10, (**e**) bottom surface of PU15-10, (**f**) cross-section of PU15-10.

**Figure 5 membranes-10-00008-f005:**
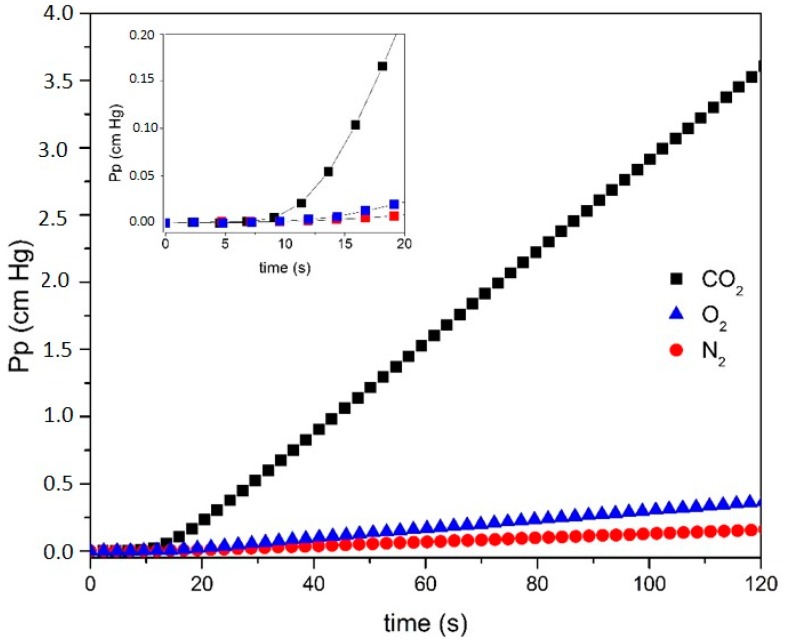
Permeate pressure (*Pp*) vs. time (*t*) of CO_2_ at Pf = 269 cmHg (black square), O_2_ at *Pf* = 235 cmHg (blue triangle) and N_2_ at *Pf* = 270 cmHg (red circle) for the PU10 membrane. The inset shows the permeation curves in the region of low permeation times (<20 s).

**Figure 6 membranes-10-00008-f006:**
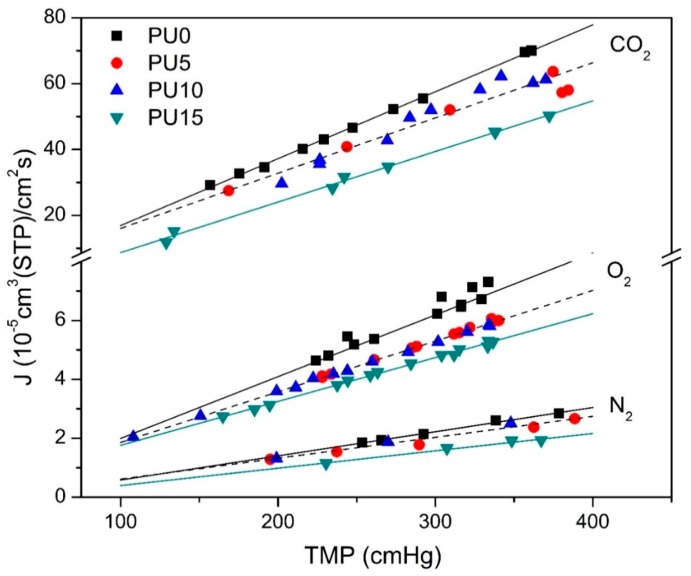
N_2_, O_2_, and CO_2_ volumetric fluxes (*J*) vs. the transmembrane pressure (TMP) for PU0 (black square), PU5 (red circle), PU10 (blue triangle), and PU15 (green inverted triangle) membranes; lines (straight-line fit using the method of least squares) PU0 in solid black, PU15 in solid green and a joint fit of PU5 and PU10 in dashed black.

**Figure 7 membranes-10-00008-f007:**
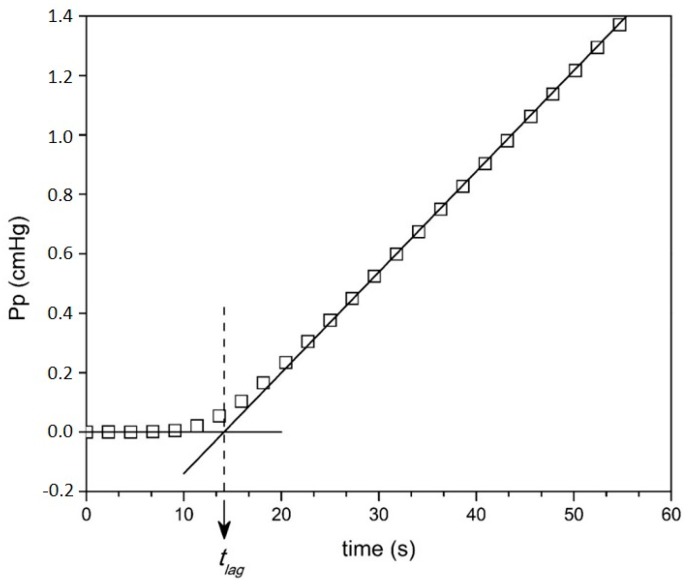
Permeate pressure vs. time for the PU10 membrane for CO_2_ at feed pressure of 269 cmHg (*Pf*). Vertical dashed line intercepts the *x* (*Pp* = 0) axis at the time lag value (*t_lag_*).

**Figure 8 membranes-10-00008-f008:**
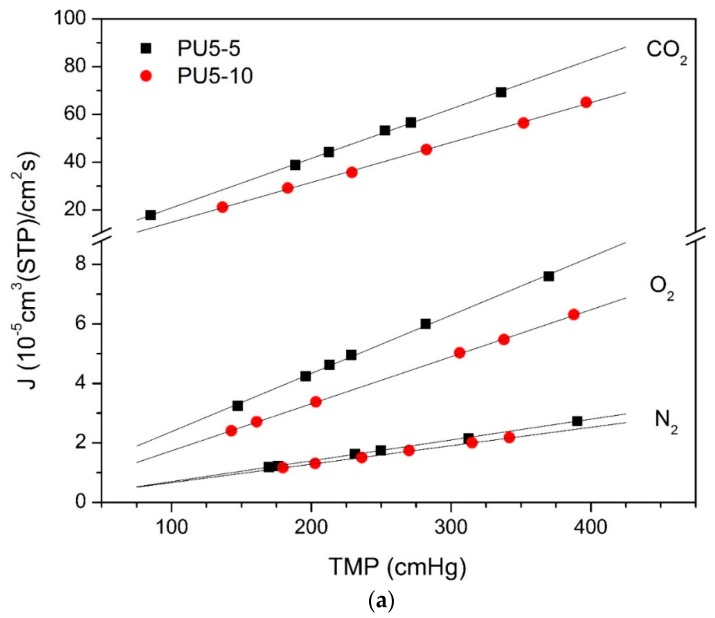
Volumetric flux (*J*) vs. TMP for the integral asymmetric membranes from group 2. Black squares represent data for membranes prepared with 5 min of solvent evaporation time and red circles data for those of 10 min. The lines are least squares fits of the points. In (**a**) data for the membranes containing 5 wt% PCL, in (**b**) data for the membranes with 10 wt% PCL, in (**c**) data for the membranes with 15 wt% PCL.

**Figure 9 membranes-10-00008-f009:**
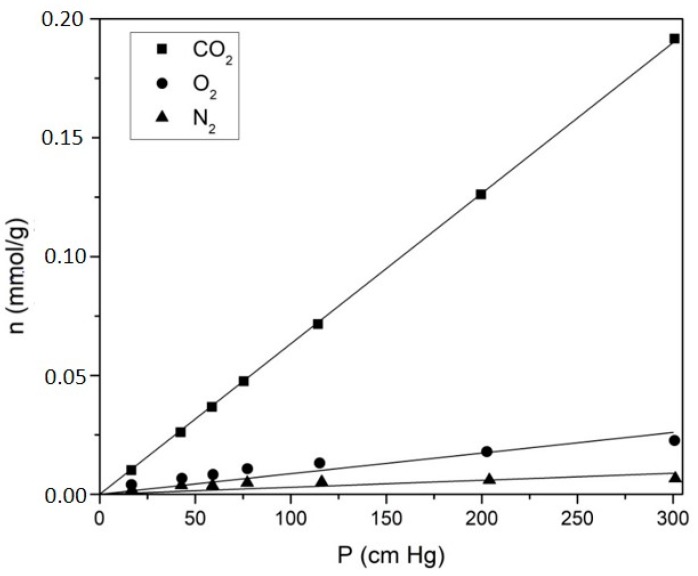
N_2_, O_2_, and CO_2_ sorption isotherms obtained by the barometric method for the PU10-1 membrane.

**Figure 10 membranes-10-00008-f010:**
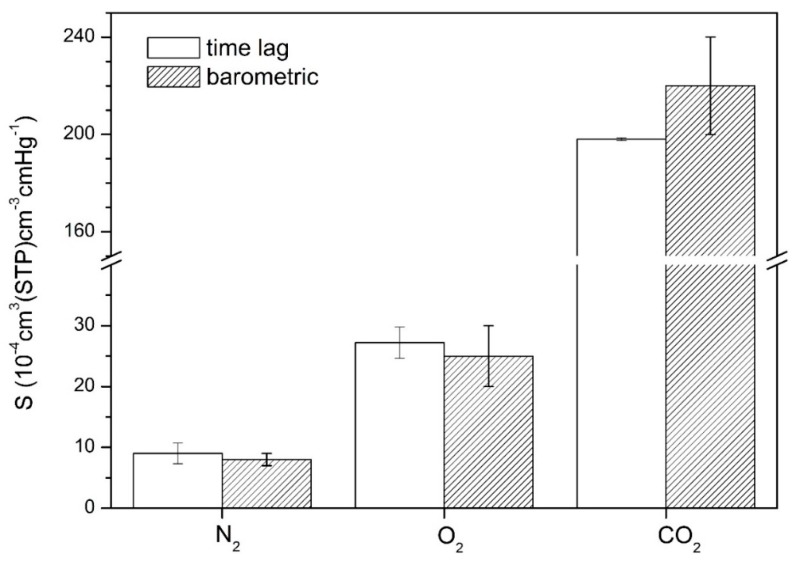
Solubility coefficients for N_2_, O_2_, and CO_2_ (average ± SD) obtained for the test membrane by the time lag method (orange) and the barometric method (blue).

**Table 1 membranes-10-00008-t001:** Thickness (l) (average ± standard deviation) of the nonporous symmetric PU membranes from group 1 and integral asymmetric PU membranes from group 2.

-	Membrane	Thickness, l (µm)
Group 1	PU0	112 ± 6
	PU5	115 ± 4
	PU10	107 ± 4
	PU15	112 ± 1
Group 2	PU5-5	107 ± 4
	PU5-10	121 ± 2
	PU10-5	109 ± 4
	PU19-10	121 ± 2
	PU15-5	103 ± 5
	PU15-10	110 ± 3

**Table 2 membranes-10-00008-t002:** N_2_, O_2_, and CO_2_ permeances and permeability coefficients (*P*) for the nonporous symmetric membranes PU0, PU5, PU10, and PU15.

-	-	$\mathrm{Permeance}\;\left(\frac{10^{-5}cm^{3}\left(STP\right)}{cm^{2}\;s\;cmHg}\right)$			*P* (Barrer)		
-	Membrane	N_2_	O_2_	CO_2_	N_2_	O_2_	CO_2_
Group 1	PU0	0.0082	0.0218	0.203	9 ± 0.5	24 ± 1.3	227 ± 12.2
	PU5 and PU10	0.0071	0.0171	0.168	8 ± 0.3	20 ± 0.7	193 ± 6.7
	PU15	0.0059	0.0149	0.154	7 ± 0.1	17 ± 0.1	172 ± 1.5

**Table 3 membranes-10-00008-t003:** Time lag (*tlag*) values, diffusion coefficients (*D*), and solubility coefficients (*S*) obtained from Table 2. O_2_ and CO_2_ permeation curves for the PU0, PU5, PU10, and PU15 nonporous symmetric membranes.

-	N_2_			O_2_			CO_2_		
-	*t_lag_*	*D*	*S*	*t_lag_*	*D*	*S*	*t_lag_*	*D*	*S*
-	(s)	$\frac{10^{-6}cm^{2}}{s}$	$\frac{10^{-4}cm^{3}}{cm^{3}cmHg}$	(s)	$\frac{10^{-6}cm^{2}}{s}$	$\frac{10^{-4}cm^{3}}{cm^{3}cmHg}$	(s)	$\frac{10^{-6}cm^{2}}{s}$	$\frac{10^{-4}cm^{3}}{cm^{3}cmHg}$
PU0	17.3	1.4 ± 0.1	7.1 ± 0.2	10.1	2.1 ± 0.4	11.3 ± 1.7	11.4	1.7 ± 0.2	140.0 ± 12
PU5	17.8	1.2 ± 0.1	6.1 ± 0.6	11.4	1.9 ± 0.2	10.6 ± 0.9	14.0	1.6 ± 0.2	123.2 ± 13
PU10	18.3	1.2 ± 0.1	8.4 ± 1.0	10.1	1.6 ± 0.2	11.9 ± 2.1	13.5	1.4 ± 0.1	160.0 ± 9
PU15	21.1	1.0 ± 0.0	6.7 ± 0.2	11.6	1.7 ± 0.1	9.6 ± 1.2	14.9	1.4 ± 0.1	119.2 ± 7

**Table 4 membranes-10-00008-t004:** N_2_, O_2_, and CO_2_ kinetic diameters and boiling points [40].

Molecule	Kinetic Diameter (Å)	Boiling Point (°C)
CO_2_	3.30	−78.5
O_2_	3.46	−183
N_2_	3.64	−196

**Table 5 membranes-10-00008-t005:** N_2_, O_2_, and CO_2_ permeances of the integral asymmetric membranes from group 2: PU5-5, PU5-10, PU10-5, PU10-10, PU15-5, and PU15-10.

-	-	$\mathrm{Permeance}\;\left(\frac{10^{-5}cm^{3}\left(STP\right)}{cm^{2}\;s\;cmHg}\right)$		
	Membrane	N_2_	O_2_	CO_2_
Group 2	PU5-5	0.0070	0.0196	0.2070
	PU5-10	0.0062	0.0158	0.1671
	PU10-5	0.0067	0.0181	0.1840
	PU10-10	0.0053	0.0146	0.1565
	PU15-5	0.0060	0.0168	0.1855
	PU15-10	0.0055	0.0153	0.1582

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
