# Peer review of "Sorption/Diffusion Contributions to the Gas Permeation Properties of Bi-Soft Segment Polyurethane/Polycaprolactone Membranes for Membrane Blood Oxygenators"

_membranes, 2020, doi:10.3390/membranes10010008_

Round 1

Reviewer 1 Report

Well-written and interesting article. A few minor text corrections in the text:

Line 92: number of reference is missing

Line 93: "affects" instead of "effects"

Lines 189, 191, 203: check if equations are correctly referred to

Line: 453: x axis instead of xx axis

Lin 551: Figure 10 instead of Figure 9

Author Response

We are very grateful for the review and for the suggested corrections, which we have incorporated into the manuscript.

Reviewer 2 Report

This paper reported a very thorough study of the membrane based blood oxygenators, and the author studied the effect of membrane structure, membrane pore size and composition on the gas permeability. All CO2, O2 and N2 have been studied and the mechanism has been well discussed. I found this paper is interesting to read and the content is useful for the membrane researchers. I would recommend its publication after some minor revisions;

It is advisable to put permeability in Table 5 also for better comparison. It is possible to discuss the stability and hemocompatible of these membranes?

Author Response

We are very grateful for the review and for the suggestions.

Regarding the first comment, we have chosen to include only the permeances in table 5, due to the difficulty of accurately measuring the thickness of the active layer of the asymmetric membranes. We thus believe that, in this case, the permeances are therefore more meaningful than the permeabilities, and we have included a paragraph about this on the manuscript.

The hemocompatibility of the PU/PCL membranes has been the subject of previous studies by our group. A more detailed mention of these studies has now been included in the introduction of the manuscript.

A detailed thorough study of the stability of the present membranes in a long time range was out of the scope of this work. However, no alteration of the properties of the membranes was observed during the time interval between synthesis and characterization experiments (up to 1 month). A paragraph with this statement has been included in the manuscript.